# Comprehensive Analysis of Highbush Blueberry Plants Propagated In Vitro and Conventionally

**DOI:** 10.3390/ijms25010544

**Published:** 2023-12-30

**Authors:** Marzena Mazurek, Aleksandra Siekierzyńska, Tomasz Piechowiak, Anna Spinardi, Wojciech Litwińczuk

**Affiliations:** 1Department of Physiology and Plant Biotechnology, Institute of Agricultural Sciences, Environment Management and Protection University of Rzeszow, Ćwiklińskiej 2, 35-601 Rzeszow, Poland; asiekierzynska@ur.edu.pl (A.S.);; 2Department of Chemistry and Food Toxicology, Institute of Food Technology and Nutrition, University of Rzeszow, St. Cwiklinskiej 1a, 35-601 Rzeszow, Poland; tpiechowiak@ur.edu.pl; 3Department of Agricultural and Environmental Sciences—Production, Landscape, Agroenergy, Università degli Studi di Milano, 20133 Milan, Italy

**Keywords:** plant in vitro culture, somaclonal variation, *Vaccinium* sp., DNA methylation, antioxidant

## Abstract

In vitro culture allows the production of numerous plants with both desirable and undesirable traits. To investigate the impact of the propagation method on highbush blueberry plants, an analysis was performed on four groups of differentially propagated plants: in vitro with axillary (TC-Ax) or adventitious shoots (TC-Ad), conventionally (SC) and using a mixed method (TC/SC). The analysis included plant features (shoot length and branching, chlorophyll and fluorescence and DNA methylation) and fruit properties (antioxidant compounds). The data obtained indicated significant differences between plants propagated conventionally and in vitro, as well as variations among plants derived from in vitro cultures with different types of explants. SC plants generally exhibited the lowest values of morphological and physiological parameters but produced fruits richest in antioxidant compounds. TC/SC plants were dominant in length, branching and fluorescence. Conversely, TC-Ax plants produced fruits with the lowest levels of antioxidant compounds. The methylation-sensitive amplified polymorphism (MSAP) technique was employed to detect molecular differences. TC-Ad plants showed the highest methylation level, whereas SC plants had the lowest. The overall methylation level varied among differentially propagated plants. It can be speculated that the differences among the analysed plants may be attributed to variations in DNA methylation.

## 1. Introduction

Blueberries are referred to as ‘superfruits’ due to their health-promoting effects. Fruits of highbush blueberries contain high levels of water (84%) and carbohydrates (9.7%) and low concentrations of proteins (0.6%) and fats (0.4%). Moreover, they are also characterised by high vitamin C content at approx. 10 mg/100 g fresh weight, corresponding to one-third of the recommended daily intake [1]. *Vaccinium corymbosum* L. is one of the fruits with the highest antioxidant potential due to its high level of polyphenols [2]. The main group of phenolic compounds in blueberries is flavonoids. This group includes anthocyanins, proanthocyanidins, as well as flavonols (mainly quercetin derivatives). The present phenolic acids mainly include chlorogenic, coumaric and ellagic acids [3,4]. The content of polyphenols in berry plants ranges from 48 to 304 mg per 100 g fresh weight. It strictly depends on the variety, growing conditions and fruit ripening. The most common anthocyanins are malvidin, delphinidin, petunidin, cyanidin and peonidin, and they occur in combination with glucose, galactose and arabinose sugar molecules. The content of anthocyanins in the fruits of *Vaccinium* plants ranges from 25 to 495 mg/100 g of fresh fruit [1]. Metabolites contained in the fruits of *Vaccinium* sp. exhibit health-promoting [5,6,7,8], anti-inflammatory and anticancer effects [5,9,10,11]. Regular consumption of blueberries is known to contribute to disease prevention [1,7,9,12].

Due to the taste and attractiveness of berries, the highbush blueberry quickly became an object of widespread interest among consumers, growers and fruit nurserymen [13]. Propagation of blueberries by conventional means can be performed by generative or vegetative methods. Generative reproduction has not been widely used in the cultivation of *Vaccinium* plants due to low seed yield, poor germination and low quality of seedlings [14,15] Vegetative propagation of blueberry plants relies on the use of semi-woody shoots collected from the donor plant, which are subsequently rooted under high-humidity conditions [16]. Technological progress and the development of in vitro culture techniques resulted in the production of berry plants, including highbush blueberries, under laboratory conditions [15,16]. The first in vitro cultures of highbush blueberry were initiated in 1979–1980 by Cohen and Elliot [17] and Zimmerman and Broome [18]. Currently, the in vitro culture method is the most widely used technique in plant biotechnology [19].

Both micropropagation and conventional plant propagation exhibit advantages and disadvantages. The main advantage of conventional methods of vegetative propagation of blueberry plants is the possibility of obtaining plant seedlings that are homogeneous with the mother plants. However, this method is slow and labour-intensive [15,19]. Other drawbacks include dependence on weather and a limited supply of seedlings dependent on the number of mother plants. These disadvantages render the conventional method insufficient to meet the growing demand for highbush blueberry plants [20]. The solution to the limitations of conventional highbush blueberry seedling production lies in the in vitro technique. In vitro propagation allows for the efficient multiplication of plants, enabling the generation of a substantial number of seedlings in a relatively short time. In vitro cultures carried out in laboratory conditions are independent of weather conditions and seasons, facilitating seedling production throughout the year. The main goal of micropropagation is to produce multiple plant clones with the same characteristics as the donor plants [19,21,22,23] Micropropagation, despite many advantages, is associated with obtaining plants that are heterogeneous compared to the mother plants. This manifested itself in various phenotypes and (epi)genetic properties [24,25,26,27,28], a phenomenon known as somaclonal variation [25,26,27,29,30,31]. Somaclonal variation occurring during in vitro cultures may be caused by pre-existing variability (explant-specific variability) or variability induced by in vitro culture [32,33,34]. The in vitro culture environment may therefore have a mutagenic effect. Plants obtained from in vitro culture of callus and protoplasts and through somatic embryogenesis may also show phenotypic changes caused by alterations at the DNA level [29]. Moreover, genetic stability during in vitro culture depends on numerous factors, including the type of medium [35], type and concentration of growth regulators [36,37], culture conditions (temperature, light, etc.) [28,38], culture duration and the number of passages [37,39]. The origin of the explants also influences somaclonal variation [40,41]. According to De Klerk [42], plants developed from adventitious shoots by indirect organogenesis are the most common carriers of somaclonal variation. It is worth noting that the appearance of adventitious shoots during in vitro cultures of many plants, including highbush blueberries, is a common, difficult-to-control phenomenon [18,43,44]. Adventitious shoots develop concurrently with axillary shoots and are often in close proximity. Some of them reach the size of axillary shoots at the end of the passage, making them difficult to distinguish [44]. According to Litwińczuk [44], tolerating adventitious shoots may lead to the uncontrolled selection of mutated or epigenetically altered cultures. The effect of somaclonal variation occurring during micropropagation of blueberry plants leads to both undesirable and desirable traits. Scientific reports [15,44,45] suggest that micropropagated *Vaccinium* plants may exhibit delayed entry into the fruiting period, lower yield in the first years, smaller fruit size and reduced fruit mass. Other studies indicated a smaller number of flowers in the inflorescences and fewer inflorescences themselves, but these features may also depend on species and variety [15,46,47,48]. On the other hand, other studies [15,44] indicated that blueberry plants originating from in vitro cultures exhibited increased root production, more vigorous growth and enhanced branching compared to traditionally propagated plants. 

To identify true-to-type plants obtained after in vitro propagation, it is essential to use methods for evaluating somaclonal variation [27]. A growing number of studies indicate that the variability observed in plants regenerated through in vitro cultures is not solely attributed to genetic alteration but also to epigenetic changes. Additionally, not all genetic variations in somaclones are phenotypically expressed. These modifications can either occur in non-coding sequences or may not significantly alter the gene product [27]. To detect somaclonal variation at the epigenetic level, characterised by alterations in DNA methylation, the methylation-sensitive amplification length polymorphism technique (MSAP) is most commonly applied [27,49,50,51,52,53]. This method is based on the different sensitivities of restriction enzymes to cytosine methylation at their cleavage sites [49,54]. It permits the comparison of the DNA methylation status of different organisms based on their differential digestion patterns. A frequently employed method involves HpaII and MspI isoschizomers, both recognising the same 5′-CCGG sequence. Although HpaII cleaves hemimethylated sequences (only one DNA strand is methylated), it is most sensitive when one or both cytosines are fully methylated (both strands are methylated). In contrast, MspI cleaves at the C5mCGG site, whether hemimethylated or with both strands methylated, but does not cleave at the 5mC5mCGG or 5mCCGG sites [53]

Previous studies on the impact of micropropagation on plants of the genus *Vaccinium* have mainly focused on lowbush blueberries, with fewer investigations conducted on highbush blueberries. These studies focused on the comparison of conventionally propagated plants with plants derived from in vitro cultures, without taking into account the type of culture [15,45,46,47]. Furthermore, existing research did not analyse comprehensively the impact of propagation methods on plants and fruits, focusing on morphological [15,44,45,46,47,55] differences mainly, or less, on physiological [56], biochemical [15,46,48] or epigenetic differences [57]. Moreover, there are no scientific reports presenting analyses conducted on a wide group of differentially propagated plants. Therefore, the present study aimed to demonstrate the differences between highbush blueberry plants (‘Brigitta’ cultivar) and their fruits, propagated conventionally, in vitro (with different types of explants) and using a combination of these methods (in vitro and conventional). Differences between plants were determined at the morphological, physiological and epigenetic levels, whereas differences between fruits were determined based on analyses of bioactive compounds.

## 2. Results

The analysis demonstrated that the propagation method (in vitro or conventional) of the highbush blueberry cultivar ‘Brigitta Blue’ (*Vaccinium* × *corymbosum* L.) significantly influenced both the plants and the fruits. Differences were detected at the morphological, physiological, epigenetic and biochemical levels.

### 2.1. Plant Analysis

#### Morphological Measurements

The studied group of blueberry plants displayed significant differences in growth strength and branching. Variations were observed in the number of shoots (both main and lateral), the average length of main shoots and the maximum length of main shoots. Plants propagated in vitro (TC-Ax and TC-Ad) and those propagated in vitro first and then conventionally (TC/SC) showed higher values of the examined parameters compared to plants propagated only conventionally (SC) (Table 1).

The highest values of the analysed features were mainly recorded in TC/SC plants, including the maximum length of shoots, the number of main and lateral shoots and the total number of shoots. In contrast, SC plants consistently demonstrated the lowest values (Table 1), whereas plants obtained directly from in vitro culture (TC-Ax and TC-Ad) showed a similar number of shoots. However, TC-Ax plants surpassed TC-Ad and other groups in terms of the average length of main shoots (Table 1).

### 2.2. Chlorophyll Content and Fluorescence

The physiological measurements of chlorophyll (a and b) content and the efficiency of the light phase of photosynthesis revealed differences between individual groups of highbush blueberry plants (Figure 1 and Figure 2). Fluorometric analyses confirmed different efficiencies of the light phase of photosynthesis between the plant groups studied. Statistically significant differences were observed in minimum (F0), maximum (Fm) and variable fluorescence (Fv), as well as in the efficiency of the PSII photosystem (Fv/Fm) (Figure 1).

The collected data indicated that TC/SC highbush blueberry plants exhibited the highest fluorescence values: minimum (F0), maximum (Fm) and variable (Fv) fluorescence. Conventionally propagated blueberry plants (SC) displayed significantly lower fluorescence parameters, except for Fv/Fm. However, blueberry plants derived directly from in vitro (TC-Ax and TC-Ad) cultures demonstrated similar values of these parameters (F0, Fm and Fv). Nonetheless, TC-Ad plants indicated the lowest efficiency of the PSII photosystem (Fv/Fm) compared to TC-Ax plants, as well as other groups (TC/SC and SC). Measurements of chlorophyll a and b content revealed statistically significant differences between plants propagated using different methods (Figure 2). Plants directly derived from in vitro cultures (TC-Ax and TC-Ad) and those propagated using combined methods (TC/SC) showed similar levels of chlorophyll a, b and a + b. SC plants, on the other hand, had significantly lower chlorophyll a and a + b contents (Figure 2).

### 2.3. Fruit Analysis

Fruits were picked up from propagated in vitro and conventionally 5-year-old blueberry plants and stored at −80 °C. Homogenised fruits were used for analyses of their antioxidant potential. The result indicated differences in some parameters between in vitro and conventionally propagated blueberry plants.

#### 2.3.1. Antioxidant Activity

The value of the antioxidant activity varied between blueberry plants propagated using different methods. The highest DPPH value was observed in SC (18, 57) and TC-Ad plants (18, 35), whereas the ABTS value was higher in TC-Ad and TC/SC plants. However, a lower level of antioxidant activity (DPPH and ABTS) was recorded for TC-Ax plants (Figure 3).

#### 2.3.2. Polyphenols and Anthocyanins

The determination of polyphenol and anthocyanin contents also showed that SC plants had the highest concentrations of these compounds, while TC-Ax plants (as well as TC/SC) had their lowest concentrations (Figure 4). 

#### 2.3.3. Ascorbic Acid

The ascorbic acid content varied in blueberry fruits from plants obtained directly from in vitro culture, propagated with different types of explants (axillary or adventitious shoots). TC-Ad plants had the highest level of ascorbic acid, while TC-Ax had the lowest level. These values differed significantly from each other as well as from other groups of plants (TC/SC and SC). The analysis of the ascorbic acid levels indicated similar values of this parameter for plants propagated conventionally (SC) or using combined methodologies (in vitro propagation followed by conventional cultures) (TC/SC) (Figure 5). 

#### 2.3.4. DNA Methylation

Molecular analyses were performed using MSAP markers. This method utilises *EcoR*I/*Hp*aII and *EcoR*I/*Msp*I restriction enzymes with different sensitivity to cytosine methylation within the recognised 5′CCGG3′ DNA sequence. MSAP analysis produced polymorphic DNA fragments, distinguishing individual plant groups from each other (Figure 6). Notably, polymorphisms in DNA band patterns were also detected between plants derived from in vitro culture propagated by axillary (TC-Ax) or adventitious shoots (TC-Ad) (Figure 6B,C).

DNA fragments characterising M-type (fully/symmetric methylation of the 5′CCmGG3′ sequence) and H-type (hemimethylation of the 5′CmCGG3′ sequence) methylation events were identified according to the procedure described by Xiong et al. [54] and Walder et al. [58]. For all analysed groups of blueberry plants, MSAP analysis yielded a significantly higher number of amplification products, indicating M-type methylation sites compared to H-type methylation events (Table 2). 

Methylation frequency analysis showed different levels of methylation between blueberry plants propagated conventionally or in vitro, as well as using a combination of these methods. The highest methylation frequency was recorded for TC-Ad plants, while TC/SC plants had slightly lower levels of methylation. On the other hand, the lowest occurrence of methylation events was found in traditionally propagated SC plants and in vitro-derived plants (TC-Ax) (Table 2). Interestingly, plants from in vitro culture, propagated through axillary or adventitious shoots, demonstrated different methylation frequencies (Table 2). These findings underscore the presence of somaclonal variation during in vitro culture.

## 3. Discussion

Historically, the primary objective of micropropagation was to obtain true-to-type plants faithfully replicating the characteristics of the donor plant in a relatively short time. In scientific terms, in vitro culture methods for plants were generally regarded as a means of cloning a specific genotype [23,28,57]. Consequently, in breeding programmes, plants with desirable agronomic traits were selected for micropropagation to quickly obtain clones with the same properties (i.e., plants morphologically and genetically identical to the mother plant) [22,23,59].

Compared to the conventional propagation method, micropropagation is a more effective, reproducible and season-independent approach [21,23]. As a result, micropropagation has rapidly become, if not the primary technique, a prominent method for propagating large numbers of plants, including those belonging to the genus *Vaccinium* sp. [43,57].

For most nurserymen and cultivators, using the vegetative propagation method is important to preserve key agronomic features that characterise an elite variety [22,59]. However, the occurrence of somaclonal variation makes it difficult to meet this requirement. Uncontrolled changes during in vitro cultures can result in different phenotypes with both undesirable and improved features [30]. Nevertheless, when in vitro cultures are used to clone true-to-type plants with specific and selected characteristics, any changes at the morphological and/or molecular level are not desirable [29]. Regarding micropropagated blueberry plants, it has been proven so far that emerging undesirable traits include lower yield, delayed fruiting, smaller fruit size and fewer flowers on the inflorescence [15,45].

The present study has confirmed the hypothesis that the method of plant propagation affects both the plants and the fruits of highbush blueberries. The results indicated differences among plants propagated conventionally (by semi-woody cuttings; SC plants) and those propagated through the in vitro method using various explants (axillary shoots, TC-Ax, or adventitious shoots, TC-Ad), as well as plants propagated through a combined approaches (in vitro followed by conventional propagation; TC/SC plants). Variations between the analysed plant groups were detected at the morphological, physiological, epigenetic and biochemical levels. Specifically, SC plants exhibited notably lower values in terms of the number of main and total shoots (Table 1) and chlorophyll a and a + b contents (Figure 2) compared to other groups. Conversely, TC/SC plants displayed the highest values for parameters such as the maximum length of main shoots and the number of total and lateral shoots (Table 1). TC-Ax plants demonstrated the statistically significantly highest value for the average length of main shoots. Highbush blueberry plants directly originating from in vitro cultures (propagated by axillary or adventitious shoots) generally displayed intermediate values of the analysed parameters when compared to SC and TC/SC plants. Notably, differences between plants propagated through in vitro culture and those propagated conventionally were also previously observed by Goali et al. [46] and Litwińczuk [44]. Goyali et al. [46] highlighted a significant interaction between the propagation method, plant branching and chlorophyll content in lowbush blueberries. Similar observations were reported by Litwińczuk et al. [55]. The latter authors noted that highbush blueberry plants derived from softwood cuttings exhibited slower growth, produced significantly fewer and shorter shoots and showed greater variability compared to micropropagated plants. Such differences between cutting-derived and micropropagated blueberry plants were also observed earlier by Grout et al. [60] and El-Shiekh et al. [61]. The aforementioned differences could be the result of the juvenile characteristics of micropropagated plants [43,57], which may persist in the long term and may facilitate rapid establishment of micropropagated highbush blueberry plants in a new planting area. According to El-Shiekh et al. [61], the enhanced branching and spreading characteristics of tissue culture-derived blueberry plants can persist for up to 10 years. Scientific reports indicate that plants obtained by micropropagation manifested juvenile features, including high rooting capacity of shoots and intensive vegetative growth [43,57]. Research performed on *Vaccinium* sp. plants has validated these phenomena in highbush and lowbush blueberries [15,43,44,45,46,47,55], lingonberries [48,62,63] and cranberries [64,65]. However, most authors of the cited studies did not consider the origin of the cultures (derived from axillary or adventitious shoots) and focused solely on comparing plants from in vitro cultures to conventionally propagated ones.

In the present study, we demonstrated variations in chlorophyll content and fluorescence between plants derived from in vitro cultures and those propagated conventionally. Differences were also observed in the reaction to stress conditions of highbush blueberry plants subjected to diverse propagation methods [56]. Many authors [66,67,68] have indicated that in vitro culture conditions may affect photosynthetic processes. Some researchers have observed that in vitro conditions induce alterations in the shape of chloroplasts, starch accumulation or irregular orientation of the thylakoid system [68,69]. Therefore, such abnormalities are believed to be responsible for the reduced photosynthetic efficiency of regenerated plants [70]. The diverse fluorescence values obtained in this study for micropropagated plants and traditionally propagated plants may be due to the application of growth regulators during in vitro cultures [57]. This notion is supported by the studies of Stefanova et al. [68] and Dobránszki and Drienyovszki [66], who demonstrated that growth regulators could modify the morphology and anatomy as well as the functioning of the photosynthetic apparatus.

Analysis of antioxidant parameters of fruits, such as antioxidant activity and levels of ascorbic acid, total polyphenols and anthocyanins, also revealed differences between highbush blueberry plants propagated conventionally and by using in vitro methods (Figure 3, Figure 4 and Figure 5). Interestingly, conventionally propagated blueberry plants (SC) frequently demonstrated superior values of the analysed parameters (total polyphenolics and anthocyanins and antioxidant activity—DPPH activity) (Figure 3 and Figure 4) compared to other groups of plants. Equally high (total anthocyanins and antioxidant activity—DPPH activity) or even higher values (ascorbic acid level—ABTS activity) were recorded for blueberry plants derived from in vitro cultures using adventitious shoots as explants (TC-Ad) (Figure 3 and Figure 4). It is noteworthy that plants propagated in vitro by axillary shoots (TC-Ax) as explants demonstrated the lowest values in almost all analysed parameters (antioxidant activity, total polyphenol and anthocyanin contents, as well as ascorbic acid levels) (Figure 3, Figure 4 and Figure 5). Goyali and colleagues published two reports in 2013 [46] and 2015 [15], providing evidence for a strong and positive correlation between total phenolic and anthocyanin contents and antioxidant activity in blueberries.

There are numerous literature reports highlighting the influence of various factors on the quality of blueberry fruit. For *Vaccinium* plants, the content of bioactive compounds, such as vitamin C, sugars and other components like minerals, depends on geographical location, soil, climatic and habitat conditions, crop care and harvest date [16]. However, there is limited scientific literature addressing the impact of the propagation methods of blueberry plants on the content of bioactive compounds. Goyali et al. [46] indicated that the propagation method exerted a significant effect on total polyphenols in lowbush blueberries. However, the next detailed analyses performed by the same group [15] demonstrated that micropropagated lowbush blueberry plants were characterised by a higher content of polyphenols and flavonoids compared to traditionally propagated plants [28,71]. The latter findings contradict the results obtained in the presented research for highbush blueberry plants, which can be attributed to different combinations and concentrations of plant growth regulators (PGR) applied during in vitro cultures. Various types and concentrations of PGRs have been reported to exert diverse regulatory effects on developmental processes and concentrations of secondary metabolites in plants [72,73]. Hormone concentration is the major factor in secondary product accumulation, such as phenolics and flavonoids [74,75,76]. Plant hormones are chemical compounds and a group of key signal molecules that are actively involved in the synthesis of plant secondary metabolites and also in regulating development and plant growth [74,75,76]. According to Baskaran et al. [73], a combination of glutamine and N6-benzyladenine significantly increased the accumulation of phenolics and flavonoids in vitro compared to the separate application of these compounds. In our study, only cytokinins, specifically 2iP (6-γ,γ-dimethylallylamine) at a concentration of 10 mg/L, were used in the medium. Research performed by Al-Khayri et al. [74] indicated that cell suspension cultures of date palm (*Phoenix dactylifera* L.), containing 2,4-D and 2iP, yielded the maximum accumulation of phenolics only in the case when 2iP is combined with 2,5-D at a concentration of 2.5 mg/L (2iP) and 5 mg/L (2,4-D). Whereas the cell suspension culture medium supplemented with a higher concentration of auxin/cytokinin (10 mg/L 2,4-D + 5 mg/L 2iP) led to the accumulation of the least concentration of the total phenolic content and flavonoids of date palm. On the other hand, Bairu et al. [29] and Vitamvas et al. [77] have documented a clear association between culture type, in vitro morphogenesis and the incidence of somaclonal variation. Leva et al. [31] categorised cultures based on genetic stability, ranking those established by explants with pre-formed apical and lateral meristems, that is, growth tips and nodal shoot sections, as the most stable. Following these were adventitious shoot cultures started from de novo meristems, particularly through direct organogenesis, somatic embryo cultures developed by direct embryogenesis, organ cultures regenerated from callus (direct morphogenesis), suspension cultures (of cells) and protoplast cultures. The appearance of callus and adventitious shoots during blueberry in vitro cultures is usually considered a potential source of somaclonal variation [44]. Comparing TC-Ax plants to TC-Ad ones, we showed statistically significant differences for parameters such as PSII photosystem efficiency (Fv/Fm) and the average length of main shoots. These parameters reached higher values for TC-Ax plants compared to TC-Ad and SC plants (average main shoot length). 

The conducted research revealed that plants derived from tissue cultures and propagated by axillary shoots (TC-Ax plants) exhibited more significant differences from conventionally propagated plants (SC) than those propagated by adventitious shoots (TC-Ad). This observation is intriguing, as adventitious shoot cultures are commonly considered to be a source of somaclonal variation. Perhaps the researchers’ focus on finding evidence of somaclonal variability in plants obtained from adventitious shoots, generally considered as a source of somaclonal variation, was misplaced. Debnath [78], in his molecular analyses of plants obtained from adventitious shoots, did not include plants obtained from axillary shoots (commonly regarded as being genetically stable). Soneji et al. [79], on the other hand, observed phenotypic differences in pineapple micropropagation using axillary buds, particularly in fruit colour and thorn generation [79]. According to the aforementioned authors, cultures derived from axillary shoots are the source of somaclonal variation. The reason for differences between plants propagated in in vitro cultures may be attributed to changes occurring at the genetic and/or epigenetic level.

Molecular analyses conducted by researchers focusing on determining the clonal identity of plants of the genus *Vaccinium* confirmed the absence of differences between clones of adventitious origin of lowbush blueberry [78], as well as between cultures and mother plants of highbush blueberry [80]. However, it is crucial to note that these analyses applied EST-PCR [78] and RAPD [80] molecular markers. These techniques are applied to detect alterations in DNA sequence; epigenetic analyses, on the other hand, are designed to detect changes in DNA structure, encompassing mechanisms such as gene silencing or activation, often influenced by changes in DNA methylation patterns [28,49,81]. Therefore, the methylation process plays a pivotal role in gene expression in eukaryotes, influencing various aspects of plant growth and development as well as responses to stress [28,50,81]. In vitro cultures create conditions conducive to changes in gene expression. Hence, it can be inferred that obtaining micropropagated plants different from the initial plant may not necessarily involve mutations but could result from a modified DNA methylation profile [57].

Epigenetic analyses based on the determination of DNA methylation using MSAP markers showed different methylation levels among the analysed groups of differentially propagated highbush blueberry plants (Table 2). A prevalence of symmetric or fully methylated 5′CmCGG 3′ DNA sequence (Type M) over hemimethylated 5′mCCGG 3′ DNA sequence (Type H) was found in all plant groups (Table 2). This predominance of total (symmetric) cytosine methylation over hemimethylation has been previously confirmed in other scientific reports [51,52,53]. Goyali et al. [51] showed a higher frequency of symmetric methylation events compared to hemimethylated events in both conventionally and in vitro propagated lowbush blueberry plants. However, the highest degree of methylation (frequency) was found in TC-Ad plants, slightly lower in TC/SC plants and lowest in SC plants and TC-Ax (Table 2). 

DNA methylation has been recognised as a major regulatory epigenetic mechanism, closely associated with diverse gene functions [51]. Epigenetic control of gene expression plays a fundamental role in cell division and differentiation, influencing various functions [51,82]. This regulatory system of gene expression is related to developmental stage, tissue type, environmental conditions and vigour [83]. Changes in methylation patterns are therefore fundamental to plant development processes [84]. Consequently, it can be hypothesised that the observed differences between the plants propagated conventionally (SC), using in vitro techniques (TC-Ad and TC-Ax), as well as by a combination of these approaches (TC/SC), at the morphological, physiological and biochemical levels, may be attributed to variations in DNA methylation levels.

## 4. Materials and Methods

The experiments included four groups of nursery plants of the highbush blueberry (*Vaccinium* × *corymbosum* L.) cv. ‘Brigitta Blue’. Two groups were obtained from in vitro cultures propagated through axillary (TC-Ax) or adventitious shoots (TC-Ad). The control group of plants (SC) was obtained by rooting semi-woody cuttings derived from continuously conventionally propagated plants. An additional fourth group (TC/SC) was also included, consisting of plants obtained by rooting semi-woody cuttings from conventionally propagated plants originally derived from in vitro cultures. All groups contained approximately 40 plants.

### 4.1. In Vitro Conditions

For micropropagation, a modified ZB medium (Z-2) [18] was used with the addition of L-cysteine (5 mg/L), vitamin C (100 mg/L), sucrose (25 g/L) and fructose (5 g/L). The medium was supplemented with the following growth regulators: cytokinins (10 mg/L 2iP; 6-γ,γ-dimethylallylamine), adenine sulphate (80 mg/L) and auxin (4 mg/L IAA; indolyl-3-acetic acid). Stabilised highbush blueberry cultures, maintained in in vitro culture for 3 years, were used as the starting material for TC-Ax and TC-AD in vitro cultures. The donor plant material for the initiation of the in vitro stabilised culture consisted of one-year-old nursery pot plants grown in a glasshouse. They were obtained conventionally by rooting cuttings collected from mother plants in the previous year. The mother plants were propagated in the same manner for at least 2 generations. The explants for in vitro culture initiation were prepared from new shoots/accretions taken from mother plants grown in a glasshouse. Two types of in vitro cultures were established from 2-node shoot explants of different origins. For the TC-Ax in vitro culture, a 2-node axillary shoot fragment, strongly connected with the donor culture (by vascular bundles), was used as an explant. Axillary shoots were developed from a bud formed in the leaf axils. For the TC-Ad in vitro culture, a 2-node adventitious shoot fragment that had developed from a callus or was weakly connected with the donor culture (without vascular bundles) was used as an explant. Examples of explants and in vitro cultures are included in Appendix A. The highbush blueberry cultures were maintained for 10–12 weeks and then passaged to fresh ZB medium. Four passages were performed. Plant cultures were maintained under a 16-h photoperiod, with light provided by cool-white fluorescent lamps (photosynthetic photon flux density (PPF) of approximately 12 µmol m^−2^ s^−1^) at 24 ± 2 °C.

### 4.2. In Vivo and Field Conditions

All plant groups were obtained by in vivo rooting of 3-node shoot fragments derived from donor plants (SC and TC/SC) or in vitro cultures (TC-Ax and TC-Ad). Rooting was carried out in ‘mini-greenhouses’ under high relative air humidity at 26 °C. The substrate used for rooting was a mixture of peat and perlite in a ratio of 3:1 (*v*/*v*).

The rooted shoots were placed in pots (9 x9 cm, initially with a capacity of 0.5 L and 1.5 L afterwards). After three years, the plants were transplanted into the ground. The blueberry plants grew in a substrate prepared from a mixture of high peat with a pH of 3.5–4.0 and sand in a ratio of 2:1 (*v*/*v*). The plants were watered at 1–3 day intervals (depending on climatic conditions) and fertilised from April to September with a 0.3% solution of ‘Florovit’ (IncoVeritas, Góra Kalwaria, Poland) containing the macronutrients (N, K and P (4:6:6)) and micronutrients( B, Cu, Fe, Mn and Zn). The plants grew under field conditions in a sunny location protected from the wind in south-eastern Poland, in Karolówka, Podkarpackie Voivodeship.

The analyses were performed in the following periods: 2014—obtaining two types of in vitro culture of highbush blueberries, TC-Ax and TC-AD, and passaging to fresh medium; 2015—rooting of blueberry shoots derived from donor plants; 2018—morphological, physiological and epigenetic analyses; 2020—fruit collection; 2023—analysis of bioactive compounds.

### 4.3. Plant Analysis

#### 4.3.1. Morphological Measurements

Morphological analysis was performed on 3-year-old seedlings before transplanting them from pots to the ground. Measurements included the length of each shoot and the total number of shoots, with only those exceeding a minimum length of 3 cm taken into account.

#### 4.3.2. Chlorophyll Content and Fluorescence

An IMAGING-PAM M-Series Chlorophyll Fluorometer (MINI version) (Waltz GmbH Eichenring, 6 91090 Effeltrich, Germany) was used to record chlorophyll fluorescence parameters. The leaves were kept in the dark for at least 20 min before measurements. The analyses were performed in 4 replicates for 20 leaves from randomly selected plants from each group. The following parameters were determined: minimum fluorescence (F0) and maximum fluorescence (Fm). Based on these measurements, the variable fluorescence (Fv = Fm − F0) and PSII photosystem efficiency (Fv/Fm) were calculated.

The content of photosynthetic pigments (chlorophyll a, chlorophyll b and total a + b) was determined on 20 leaves sampled from randomly selected plants of different origins. Pigments were extracted from 50 mg of leaf tissues with 2.5 mL of DMSO (dimethyl sulfoxide, Toruń, Poland). Absorbance measurements were performed at 665 and 649 nm using a Specol Aquamate VIS device (Thermo Scientific, Waltham, MA, USA). The content of individual pigments was calculated according to the formulas given below and described earlier by [85]:-chlorophyll a = [(12.9 · A663) − (3.45 · A649)].-Chlorophyll b = [(21.91 · A649) – (5.32 · A663).]

where A is absorbance at a specific wavelength.

### 4.4. DNA Methylation

DNA methylation was determined in young, fully developed leaves of highbush blueberry plants collected from 40 plants in each analysed group. DNA was isolated from pooled samples according to the protocol of Doyle and Doyle [86]. The methylation-sensitive amplification polymorphism (MSAP) technique, based on the protocols described in [49,54], was used. The set of primers utilised for MSAP analysis is listed in Table 3. 

DNA (50 ng) from each sample was digested with EcoRI/MspI (Thermo Scientific, Waltham, MA, USA) and EcoRI/HpaII (Thermo Scientific, Waltham, MA, USA) restriction enzymes and ligated with EcoRI- and MspI- or HpaII-specific adapters (Genomed, Warsaw, Poland) at 37 °C for six hours. The ligated DNA was diluted 10 times and pre-amplified using EcoRI and MspI or HpaII primers (Genomed, Warsaw, Poland). The PCR conditions were as follows: 94 °C—1 min, 65 °C—1 min and 72 °C—1 min for 35 cycles, with a final extension at 72 °C for 7 min. The pre-amplified product was diluted ten times with Tris-EDTA (TE) buffer and selectively amplified with different combinations of EcoRI and MspI or HpaII MSAP primers (Genomed, Warsaw, Poland) (Table 3), each with two to three selective nucleotides.

For selective amplification, PCR conditions were as follows: 94 °C—1 min, 65 °C—1 min (temperature was reduced by 0.7 °C in each successive cycle) and 72 °C—1 min for 11 cycles, and 94 °C—1 min, 56 °C—1 min and 72 °C—1 min for 23 cycles, with a final extension at 72 °C for 7 min. An equal volume of formamide dye was added to the PCR products, which were subjected to electrophoretic separation on a 6% denaturing polyacrylamide gel. Gels were stained with silver nitrate (Avantor Performance Materials Poland S.A., Gliwice, Poland (according to the method described by Bassam and Gresshoff [87]) and subsequently scanned for data recording.

#### Methylation Analysis

Methylation analysis was performed according to the method outlined by Walder [58] and Xiong et al. [54]. The presence of bands from the EcoRI + MspI (M) primer mixture and their simultaneous absence in the EcoR I + HpaII (H) reaction indicated DNA methylation. The internal cytosine of the 5′CCGG 3′ sequence was methylated (5′CmCGG 3′). This is referred to as ‘symmetric or full methylation’. The absence of bands from the EcoRI + MspI (M) reaction and their simultaneous presence in the EcoR I + HpaII (H) reaction primer mixture indicated DNA methylation, specifically the methylation of the external cytosine of one DNA strand (5′mCCGG 3′). This phenomenon is referred to as the ‘hemimethylated state’. Methylation frequency was calculated following the method described by Li et al. [53] as follows:Methylation (%) = (number of methylated bands)/total number of bands) × 100.

### 4.5. Fruit Analysis

Fruits were harvested from each group of 5-year-old blueberry plants and stored at −80 °C until analysis. The analyses were conducted on ten fruits with similar shapes and weights in 3 replicates. Homogenised fruits were used for analyses of basic quality parameters and antioxidant potential.

#### 4.5.1. Antioxidant Activity

The sample for determining antiradical activity (AA) was prepared by homogenising the tissue (5 g) with 15 mL of 75% methanol solution and centrifuging the homogenate at 10,000× *g* for 30 min. 

The antiradical activity (AA) of fruits was assayed using synthetic ABTS•+ and DPPH• radicals, following the method described by Piechowiak et al. [3]. The results of AA were expressed as Trolox equivalents (mg TE/g FW). 

Ascorbic acid content was analysed using the 2,6-dichlorophenolindophenol/spectrophotometric method as outlined by Piechowiak et al. [4]. Ascorbic acid from the fruits was extracted by homogenising the fruit tissue (5 g) with 15 mL of 2% oxalic acid and centrifuging at 10,000× *g* for 30 min. The supernatant was used to detect the ascorbic acid content. The absorbance of the reaction mixture was measured at 500 nm, and the results were presented as ascorbic acid per 100 g fresh weight (FW) of fruit tissue. 

#### 4.5.2. Total Polyphenol and Anthocyanin Contents

Approximately 8 g of frozen berries of equal size were homogenised in an ice-cooled mortar with 20 mL of cold extraction solution EtOH/formic acid/H_2_O (25/2/73). The homogenate was placed in an ultrasonic bath for 15 min. The suspension was centrifuged at 10,000× *g* for 20 min at 4 °C, and the supernatant was collected. The pellet was resuspended in 10 mL of extraction solution and treated as described above. Then, the supernatants were combined, and the volume was adjusted to 40 mL using the extraction solution. All extracts were stored at −80 °C before spectrophotometric analysis. For each group of plants, 3 pools of berries were extracted, yielding 3 biological replicates.

The total polyphenol content was determined following the Folin-Ciocalteau method [88]. To 0.05 mL of the extract in a 10 mL volumetric flask, 0.05 mL of Folin-Ciocalteau reagent, 4.45 mL of distilled water, and 2 mL of 10% Na_2_CO_3_ were added and immediately diluted with distilled water. The optical density was measured after 90 min at 700 nm using a 10S UV–vis spectrophotometer (Thermoscientific, Waltham, MA, USA). Results were expressed as milligrams of gallic acid per 1 g fresh weight (FW).

The total anthocyanin content was determined using a 10S UV–vis spectrophotometer (Thermoscientific, Waltham, MA, USA) by measuring the absorption peak of anthocyanin pigments at 530 nm. The total anthocyanin content was expressed as cyanidin 3-glucoside equivalent using a molar extinction coefficient of 26.9 L mol^−1^ cm^−1^ and reported as milligrams per 1 g FW.

### 4.6. Statistical Analysis

Statistical analysis was performed using Statistica (Stat Soft, Krakow, Poland, 13.1 version). An ANOVA test was used to identify significant differences between groups. Tukey’s HSD post hoc test was performed to determine and verify differences at a significance level of *p* ≤ 0.05.

## 5. Conclusions

In summary, the result indicated that the method of blueberry plant propagation (conventional and in vitro) influenced plant seedlings, including shoot length and branching, chlorophyll content and fluorescence, as well as DNA methylation. Additionally, differences were observed in the concentrations of fruit antioxidant compounds, including total polyphenols and anthocyanins, ascorbic acid and antioxidant activity. These differences were primarily observed between conventionally propagated highbush blueberry plants and those propagated in vitro. However, some differences were also recorded between plants derived from in vitro cultures with axillary (Tc-Ax) or adventitious shoots (TC-Ad) used as explants. The differences between the analysed groups of plants may be the result of somaclonal variation occurring during the micropropagation process. The underlying factor for somaclonal variation could be attributed to changes in DNA methylation levels detected in the analysed plant groups. It can be hypothesised that the detected differences between conventionally and in vitro propagated highbush blueberry plants may be the result of differences in DNA methylation levels. Nevertheless, further analyses should be performed to confirm the present results for other varieties of highbush blueberry plants.

## Figures and Tables

**Figure 1 ijms-25-00544-f001:**
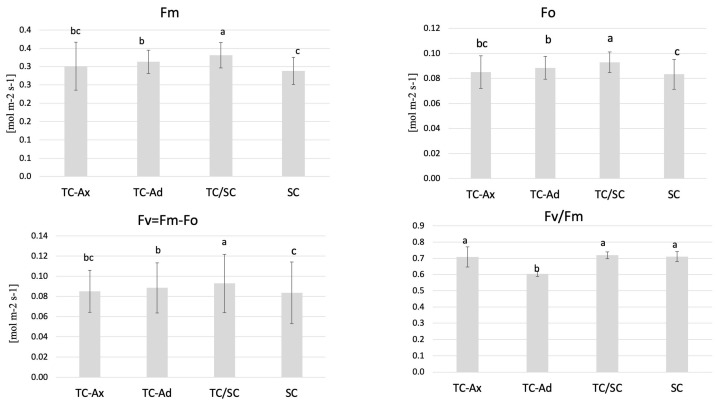
Fluorescence parameters. The mean value marked with different lowercase letters differs significantly at *p* ≤ 0.05.

**Figure 2 ijms-25-00544-f002:**
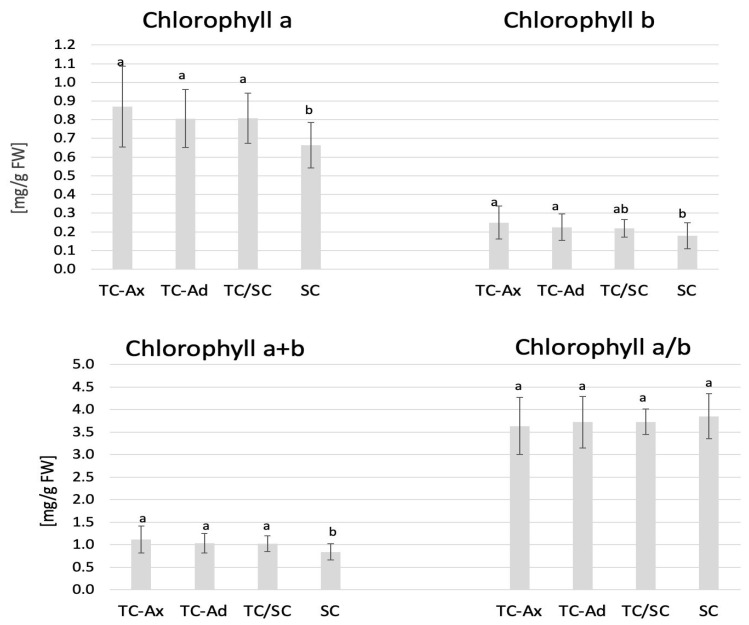
Chlorophyll content. The mean value of chlorophyll marked with different lowercase letters differs significantly at *p* ≤ 0.05.

**Figure 3 ijms-25-00544-f003:**
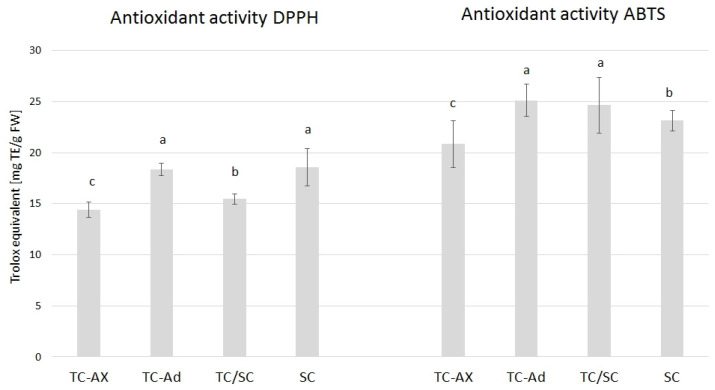
Results of antioxidant activity. The mean value marked with different lowercase letters differs significantly at *p* ≤ 0.05.

**Figure 4 ijms-25-00544-f004:**
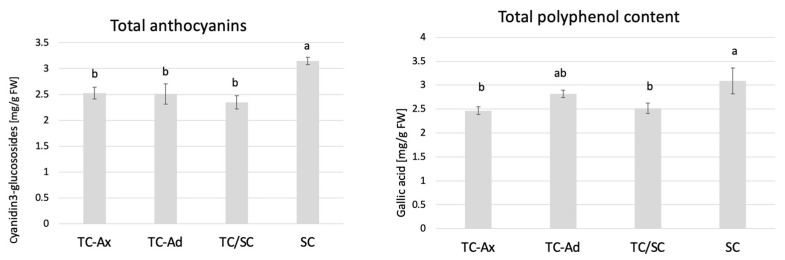
Total anthocyanins and total polyphenol contents. The mean value marked with different lowercase letters differs significantly at *p* ≤ 0.05.

**Figure 5 ijms-25-00544-f005:**
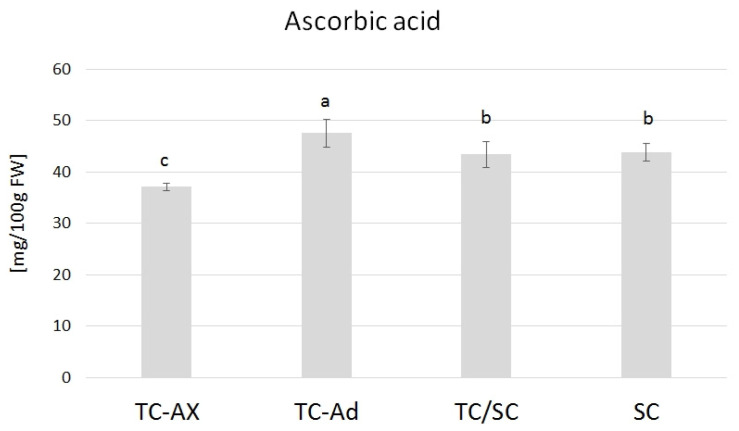
Ascorbic acid content. The mean value marked with different lowercase letters differs significantly at *p* ≤ 0.05.

**Figure 6 ijms-25-00544-f006:**
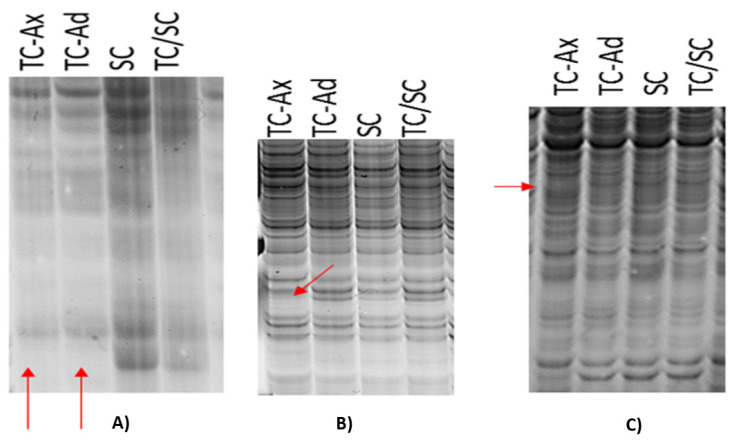
Differences in DNA band patterns—products of selective amplification as part of MSAP analysis. Red arrows indicate the lack of specific DNA bands for in vitro-derived plants (TC-Ax and TC-Ad) (**A**) or for TC-Ax plants only (**B**,**C**).

**Table 1 ijms-25-00544-t001:** Shoot measurements of highbush blueberry plants propagated by in vitro and conventional methods.

Parameters	TC-Ax	TC-Ad	TC/SC	SC
Average length of main shoots (cm)	39.4 ^a^	33.6 ^c^	37.8 ^ab^	34.5 ^bc^
Maximum length of main shoots (cm)	61.5 ^b^	57.2 ^bc^	67.3 ^a^	53.3 ^c^
Number of main shoots	5.5 ^a^	6.1 ^a^	6.1 ^a^	3.8 ^b^
Number of lateral shoots	11.9 ^b^	11.5 ^b^	16.0 ^a^	11.0 ^b^
Total number of shoots	17.5 ^b^	17.6 ^b^	22.1 ^a^	14.8 ^c^

The mean value marked with different lowercase letters differs significantly at *p* ≤ 0.05.

**Table 2 ijms-25-00544-t002:** Methylation frequency based on MSAP analysis.

Type of Propagation	TC-Ax	TC-Ad	TC/SC	SC
Types of Methylation	Methylation Frequency (%)			
M	13.2	14.3	14.0	13.0
H	10.6	10.5	10.5	10.6
M + H	23.8	24.8	24.5	23.6

**Table 3 ijms-25-00544-t003:** Sequences of primers and adapters used in MSAP analysis.

MSAP Stage	Primer/Adapter	Sequence
Ligation	EcoRI-Adapter MspI-HpaII-Adapter	5′CTCGTAGACTGCGTACC 3′
		3′CATCTGACGCATGGTTAA 5′ 5′CGACTCAGGACTCAT3′ 3′TGAGTCCTGAGTAGCAG5′
Preamplification	Pre-EcoRI	5′GACTGCGTACCAATTC 3′
	Pre-MspI-HpaII	5′GATGAGTCCTGAGTCGG 3′
Selective amplification	EcoRI-ACT	5′GACTGCGTACCAATTCACT 3′
	EcoRI-AG	5′GACTGCGTACCAATTCAG3′
	EcoRI-AC	5′GACTGCGTACCAATTCAC 3′
	EcoRI-AT	5′GACTGCGTACCAATTCAT 3′
	MspI/HpaII-ATG	5′GATGAGTCCTGAGTCGGATG3′
	MspI/HpaII-CTA	5′GATGAGTCCTGAGTCGGCTA3′
	MspI/HpaII-CTC	5′GATGAGTCCTGAGTCGGCTC3′
	MspI/HpaII-CAT	5′GATGAGTCCTGAGTCGGCAT3′
	MspI/HpaII-CT	5′GATGAGTCCTGAGTCGGCT3′
	MspI/HpaII-GT	5′GATGAGTCCTGAGTCGGGT3′
	MspI/HpaII-CA	5′GATGAGTCCTGAGTCGGCA3′

## Data Availability

The data presented in this study are available on request from the corresponding author. The data are not publicly available due to a lot of data.

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
