# Peer review of "Comprehensive Analysis of Highbush Blueberry Plants Propagated In Vitro and Conventionally"

_ijms, 2023, doi:10.3390/ijms25010544_

Round 1

Reviewer 1 Report

Comments and Suggestions for Authors

Complex analysis of differences between highbush blueberry plants and fruits as an effect of propagation

The present study aimed to indicate the impact of in vitro propagation methods and conventional one on the plants growth, morphology, physiology and molecular measurement, as well as fruits quality.

But some comments are important in this MS to be accepted by the journal:

There is no novelty, other studied foxed on the effect of propagation method (in vitro and/or ex vitro) on morphology and the antioxidant capacity of blueberry plant/fruit as Goyali et al 2013    and Goyali and Igamberdiev 2015

Only the cultivar used in the current study is different   what is the new in the current investigation? and what is the advantages of highbush blueberry that make it differ from the other types or species of berry?          Please refer to this in the text

Affiliation:   1 1 Department of Physiology and Plant Biotechnology, Institute of Institute of Agricultural Sciences, Environment Management and Protection, University of Rzeszow, ĆwikliÅ„skiej 2, 35-601 Rzeszów, Poland

Delete 1     it is repeated       delete     Institute of       it is repeated     put comma after Protection    as shown above

The numbers besides the author names should be superscript   as shown for the first author

Please check the affiliations of all authors carefully     

Keywords:  Please delete the numbers      just put    ;   between the words

1. Introduction:  It is very long try to be precise    

It should be divided into 3-4 paragraphs

The first:  about the blueberry and its nutritional and economic important

The second:  the methods of its propagation     its advantages and disadvantages

The third: the obstacles that face its propagation by tissue culture technique, and what are the weaknesses or shortcomings in the previous reports on blueberry propagation which were dealt with in the current report?

The last one is the aim of study

2. Results

Statistical analysis in the tables and figures except figure 5, is totally not correct. The highest value for each parameter should take letter a,   then b, c and so on    Please check the statistical analysis carefully

 Line 132:  what do mean?       dominant over   this expression could be used for breeding or genetic parameter        not for number of shoots?

4. Materials and Methods:

In which year this long study starts and ends?

How old are the trees? when you take the explants for in vitro propagation methods  

In which conditions these trees (the source of explants) were grown?

English language is very poor, extensive revision is needed

References:

12. Pliszka, K. Borówka Wysoka; 2002;                          It is incomplete reference

Major revision is needed

Comments on the Quality of English Language

Extensive editing of English language required

Author Response

Reviewer 1:

  1. There is no novelty, other studied foxed on the effect of propagation method (in vitro and/or ex vitro) on morphology and the antioxidant capacity of blueberry plant/fruit as Goyali et al 2013    and Goyali and Igamberdiev 2015

Only the cultivar used in the current study is different   what is the new in the current investigation? and what is the advantages of highbush blueberry that make it differ from the other types or species of berry?  Please refer to this in the text.

Answer :

According suggestion, the research problem and the advantages of the analyses performed were described in more detail.(line 134-149):

Previous studies on the impact of micropropagation on plants of the genus Vaccinium have mainly focused on lowbush blueberries, with fewer investigations conducted on highbush blueberries. These works focused on the comparison of conventionally propagated plants with plants derived from in vitro cultures, without taking into account the type of culture [15,45,46, 47]. Furthermore, existing research didn't analyze comprehensively the impact of propagation methods on plants and fruits, focusing on morphological [15,44, 45,46,47,56] differences mainly, or less on physiological [57], biochemical [15,46,48] or epigenetic differences [58]. Moreover, there are no scientific reports presenting analyses conducted on a wide group of differentially propagated plants. Therefore, the present study aimed to demonstrate the differences between highbush blueberry plants (cultivar 'Brigitta') and their fruits, propagated conventionally, in vitro (with different types of explants), and using a combination of these methods (in vitro and conventional). Differences between plants were determined at the morphological, physiological, and epigenetic levels, whereas differences between fruits were determined based on analyses of bioactive compounds.

Reviewer 1:

2 ) Affiliation:   1 1 Department of Physiology and Plant Biotechnology, Institute of Institute of Agricultural Sciences, Environment Management and Protection, University of Rzeszow, ĆwikliÅ„skiej 2, 35-601 Rzeszów, Poland

Delete 1     it is repeated       delete     Institute of       it is repeated     put comma after Protection    as shown above

The numbers besides the author names should be superscript   as shown for the first author

Please check the affiliations of all authors carefully/

Answer : It was corrected as suggested

  1. Keywords:  Please delete the numbers      just put    ;   between the words

Answer : It was corrected as suggested

  1. . Introduction:  It is very long try to be precise    

It should be divided into 3-4 paragraphs

The first:  about the blueberry and its nutritional and economic important

The second:  the methods of its propagation     its advantages and disadvantages

The third: the obstacles that face its propagation by tissue culture technique, and what are the weaknesses or shortcomings in the previous reports on blueberry propagation which were dealt with in the current report? The last one is the aim of study.

Answer:

Including the above suggestion and suggestions of other Reviewers, the introduction was rearranged and divided into 5 sections:

1) Information about the blueberries (line 34-50)

2) Methods of propagation (line 51-62)

3) Advantages and disadvantages of propagation methods (line 63-104)

4) MSAP technique (according suegstion of the Reviewer 4th) (line 105-121)

5) The aim of the study (line 122-136)

  1. Results

Statistical analysis in the tables and figures except figure 5, is totally not correct. The highest value for each parameter should take letter a,   then b, c and so on    Please check the statistical analysis carefully

Answer:

We checked the markings of statistical differences (lowercase letters a,b,c,d) and we changed in Figures as well in Table 1, the designations (a for the highest value, b- smaller, and c- the smallest ones)

 Line 132:  what do mean?       dominant over   this expression could be used for breeding or genetic parameter        not for number of shoots?

Answer:

 The word  'dominant over' was changed to 'surpassed'. The sentence after correction:

(Line 173-175): However, TC-Ax plants surpassed TC-Ad and other groups in terms of the average length of main shoots (Table 1).

  1. Materials and Methods:

In which year this long study starts and ends?

Answer:

The research was carried out in stages over several years.

( Line 480-184) The analyses were performed in the following periods: 2014 – obtaining two types of in vitro culture of highbush blueberries, TC-Ax and TC-AD, and passaging to fresh medium; 2015 – rooting of blueberry shoots derived from donor plants; 2018 – morphological, physiological and epigenetic analyses; 2020 – fruit collection; 2023 – analysis of bioactive compounds.

How old are the trees? when you take the explants for in vitro propagation methods  

In which conditions these trees (the source of explants) were grown?

Answer:

(Line 449-454) The donor plant material to initiation of the in vitro stabilized culture consisted of one year old nursery pot plants grown in a glasshouse They were obtained conventionally by rooting cuttings, collected from mother plants in the previous year. The mother plants were propagated in the same manner from at least 2 generations The explants for in vitro culture initiation were prepared from new shoots/accretions taken from mother plants grown in a glasshouse.

English language is very poor, extensive revision is needed

 Answer:

According to suggestions about poor English language and extensive revision needed, the manuscript was revisioned and corrected by professional translation expert Grzegorz Burzyński. Mr Burzyński is a Freelance translator specializing in English-Polish and Polish-English translations of biological, medical, and pharmaceutical texts. He is providing proofreading services for the Journal of Animal and Feed Sciences (JAFS)

Correction confirmation and certificates have been attached as separate documents.

References:

  1. Pliszka, K. Borówka Wysoka; 2002;                          It is incomplete reference

  Answer:

The references were corrected including the publishing Institution: Państwowe Wydawnictwo Rolnicze I Leśne.

The correct reference:

[16] Pliszka, K. Borówka Wysoka; PaÅ„stwowe Wydawnictwo Rolnicze I LeÅ›ne, Warszawa, 2002.

Reviewer 2 Report

Comments and Suggestions for Authors

The manuscript[t reports the effect of propagation method on morphological and physiological level of Vaccinium x corymbosum plants, as well the molecular level.

11)      General remark: language should be improved, the text should not be unnecessary complicated;

22)    Abstract should be re-written, the molecular analysis should be more detailed discussed;

33)      Introduction should be shortened;

Lines 98-100: “ It was proven…. Plants [14,18].” – unclear;

– what is the difference between “explants axillary” or “adventitious shoots”?

The aim of the study should be better expressed; at present form is unclear;

44)      Latin names of plant species should be written using Italic font;

55)      Results:

-          2.1.1 –  decimal numbers should be written with “.”;

-          Figure 1, Figure 2, Figure 3, Figure, Figure 2, Figure 3, Figure 4, Figure 5  – there are no values of standard deviations given;

-          “Whereas conventionally propagated blueberry plants (SC) significantly marked the smallest value of fluorescence parameters”- why this is so complicated?

-          Figure 6: “Red arrows indicate the positions of specific DNA bands characteristic for analysed groups 206 of blueberry plants.” – presence or lack of bands?

66)      Discussion:

Line 254: “or adventitious ones (TC-Ad) as explants,..” what does it mean?

-          There is no need to repeat results;

-          Lines 269-275: it should be commented why such phenomena took place, description of results of someone else is insufficient;

-          “Performed research showed that TC-Ax plants characterized more significant differences to SC plants, than TC-Ad. Perhaps the researchers' focus on finding evidence of somaclonal variability of plants obtained from adventitious shoots,…..” – unclear;

77)      Materials and Methods:

4.1. “For the TC-Ax in vitro culture, a 2-node fragment of the shoot, strongly linked with donor culture..” – unclear, what means strongly linked?

Further: “weakly linked..”?

88)      Conclusions:

“The background of somaclonal variation may be the result of DNA methylation changes, detected among an analyzed group of plants. It can be supposed that the differences at the morphological, physiological, and biochemical levels, detected among the highbush blueberry plants propagated in different ways, may be the result of differences in the DNA methylation level.” – repetition;

Comments on the Quality of English Language

General remark: language should be improved, the text should not be unnecessary complicated.

Author Response

Reviewer 2

The manuscript[t reports the effect of propagation method on morphological and physiological level of Vaccinium x corymbosum plants, as well the molecular level.

11)      General remark: language should be improved, the text should not be unnecessary complicated;

Answer:

According to suggestions, the manuscript was revisioned and corrected by professional translation expert Grzegorz Burzyński. Mr Burzyński is a Freelance translator specializing in English-Polish and Polish-English translations of biological, medical, and pharmaceutical texts. He is providing proofreading services for the Journal of Animal and Feed Sciences (JAFS)

Correction confirmation of manuscript and certificates have been attached as separate documents.

Reviewer 2

22)    Abstract should be re-written, the molecular analysis should be more detailed discussed;

Answer:

According to the suggestion, the abstract was rewritten as much as possible (200 words limited deep modification of abstract). Nonetheless, some additional information about genetic analysis was included:

(Line 28-32) The methylation sensitive amplified polymorphism (MSAP) technique was employed to detect molecular differences,. TC-Ad plants showed the highest methylation level, whereas SC plants the lowest. The overall methylation level varied among differentially propagated plants. It can be speculated that the differences among the analysed plants may be attributed to variations in DNA methylation.

Reviewer 2

33)      Introduction should be shortened;

Answer:

According to the suggestion, the introduction was modified by shortening some sentences as possible.

For example, instances of the health-promoting uses of blueberries were limited to one sentence:

Original sentence 1.:

[ Line 46- 52] ‘Daily intake of blueberries is known for preventing different diseases such as obesity diabetes circulatory system diseases, including hypertension as well as gout or chronic kidney disease [3,7]. Blueberries nutraceuticals also improve vision and liver function, brain function, bone health and increase the body's immunity [5,8–10]. Antioxidant compounds present in the berries of Vaccinium plants prevent the oxidation of cholesterol and thus reduce the risk of atherosclerosis. Blueberry nutraceuticals may also prevent neurodegenerative disorders [1,9].’

Modified/shortened sentence 1:

(Line 51-54): Metabolites contained in the fruits of Vaccinium sp. exhibit health-promoting [5,8–10], anti-inflammatory and anticancer effects [3–6]. The regular consumption of blueberries is known to contribute to disease prevention [1,3,7,9].

Original sentence 2:

( Line 82 -84 ) Moreover, genetic stability during in vitro culture depends on numerous factors: type of medium [35], type and concentration of growth regulators [36,37], culture conditions (temperature, light, etc.) [28,38], culture duration and number of passages [37,39] as well as the

Modified/shortened sentence 2:

(line 95-96 ): The origin of the explants also influences somaclonal variation  [40,41]

To shorten the introduction some sentences were removed.

For example:

( Line 66-69 ) In vitro culture techniques have been improved since the beginning of the 20th century. However, plant reproduction on a huge scale, became possible only in the 1960s, when Murashige and Skoog [17] in the USA developed the composition of the medium and conditions for clonal plant reproduction [18]

However, according to the suggestion of other Reviewers, the introduction was enriched with information about biologically active substances in highbush blueberries (suggestion of Reviewer 3) and information about MSAP techniques (Reviewer 4).

This finally led to expand the introduction. Nevertheless, the introduction was systematized and divided into sections:

1) Information about the blueberries (line 34-50)

2) Methods of propagation (line 51-62)

3) Advantages and disadvantages of propagation methods (line 63-104)

4) MSAP technique (according suegstion of the Reviewer 4th) (line 105-121)

5) The aim of the study (line 122-136)

Reviewer 2

Lines 98-100: “ It was proven…. Plants [14,18].” – unclear;

Answer:

The original unclear sentence:

‘It was proven that blueberry plants originating from in vitro cultures, more roots, stronger growth, and branching were observed compared to traditionally propagated plants [14,18]’

Was changed to: (line 101-104)

On the other hand, other studies [15,44] indicated that blueberry plants originating from in vitro cultures exhibited increased root production, more vigorous growth, and enhanced branching compared to traditionally propagated plants.

Reviewer 2

– what is the difference between “explants axillary” or “adventitious shoots”?

Answer:

To explain in detail the difference between axillary and adventitious shoot the subsequent sentence was included in the text:

(Line 455- 460) Two types of in vitro cultures were established from 2-node shoot explants of different origins. For the TC-Ax in vitro culture, a 2-node axillary shoot fragment, strongly connected with the donor culture (by vascular bundles), was used as an explant. Axillary shoots developed from a bud formed in the leaf axils. For the TC-Ad in vitro culture, a 2-node adventitious shoot fragment that had developed from a callus or was weakly connected with the donor culture (without vascular bundles) was used as an explant

Reviewer 2:

The aim of the study should be better expressed; at present form is unclear;

Answer:

According suggestion, the research problem and the validity of the research aim were described in more detail.(line 123-139):

Previous studies on the impact of micropropagation on plants of the genus Vaccinium have mainly focused on lowbush blueberries, with fewer investigations conducted on highbush blueberries. These works focused on the comparison of conventionally propagated plants with plants derived from in vitro cultures, without taking into account the type of culture [15,45,46, 47]. Furthermore, existing research didn't analyze comprehensively the impact of propagation methods on plants and fruits, focusing on morphological [15,44, 45,46,47,56] differences mainly, or less on physiological [57], biochemical [15,46,48] or epigenetic differences [58]. Moreover, there are no scientific reports presenting analyses conducted on a wide group of differentially propagated plants. Therefore, the present study aimed to demonstrate the differences between highbush blueberry plants (cultivar 'Brigitta') and their fruits, propagated conventionally, in vitro (with different types of explants), and using a combination of these methods (in vitro and conventional). Differences between plants were determined at the morphological, physiological, and epigenetic levels, whereas differences between fruits were determined based on analyses of bioactive compounds.

Reviewer 2:

44)      Latin names of plant species should be written using Italic font;

Answer:

The names of plant species were corrected to italic font

Reviewer 2:

55)      Results:

    2.1.1 –  decimal numbers should be written with “.”;

Answer: The commas were replaced with dots.

Reviewer 2:

 Figure 1, Figure 2, Figure 3, Figure, Figure 2, Figure 3, Figure 4, Figure 5  – there are no values of standard deviations given;

Answer:

The standard deviations was included in all of the Figures

Reviewer 2:

-          “Whereas conventionally propagated blueberry plants (SC) significantly marked the smallest value of fluorescence parameters”- why this is so complicated?

The original (above) sentence was changed to more clear  sentence:

(Line 174-176) Conventionally propagated blueberry plants (SC) displayed significantly lower fluorescence parameters, except for Fv/Fm. However, blueberry plants derived directly from in vitro (TC-Ax and TC-Ad) cultures demonstrated similar values of these parameters (F0, Fm, Fv).

Reviewer 2:

-          Figure 6: “Red arrows indicate the positions of specific DNA bands characteristic for analysed groups 206 of blueberry plants.” – presence or lack of bands?

Answer:

The sentence were corrected:

Red arrows indicate the lack of specific DNA bands for: in vitro derived plants (TC-Ax and TC-Ad) (A); or for TC-Ax plants only (B,C).

Reviewer 2:

66)      Discussion:

Line 254: “or adventitious ones (TC-Ad) as explants,..” what does it mean?

The obtained results indicated differences among plants propagated conventionally by rooting semi-woody cuttings (SC plants), and plants propagated in vitro method, using axillary shoots (TC-Ax) or adventitious ones (TC-Ad) as explants, and TC/SC plants, propagated in mixed manner (conventionally by rooting semi-woody cuttings, originally derived from in vitro cultures).

After corection:

Answer:

(Line 276-281 ) The present study has confirmed the hypothesis that the method of plant propagation affects both the plants and the fruits of highbush blueberries. The results indicated differences among plants propagated conventionally (by semi-woody cuttings; SC plants) and those propagated through in vitro method using various explants (axillary shoots, TC-Ax, or adventitious shoots, TC-Ad), as well as plants propagated through a combined approach (in vitro followed by conventional propagation, TC/SC plants).

Reviewer 2:

-          There is no need to repeat results;

Answer:

In the discussion part, attempts were made to refer only to the results generally. It made it possible to refer in a clear and understood manner to the results obtained by other scientists. Results presented in detail in the discussion section tried to avoid or been removed.

Reviewer 2:

-          Lines 269-275: it should be commented why such phenomena took place, description of results of someone else is insufficient;

Original sentence-> Highbush blueberry plants originated directly from in vitro cultures (propagated by axillary or adventitious shoots) were characterized in most cases, by intermediate values of the analyzed parameters, in comparison to SC and TC/SC plants. Differences between plants propagated by in vitro culture and plant propagated conventionally were also observed by Goali et al. [44], and Litwińczuk [18].

Answer:

The explanation of the above sentence can be found in the line (309-315). A paragraph was used unnecessarily (originally on line 282) which would have initially indicated the lack of reference to the above content.

Line 299-308:

The aforementioned differences could be the effect of juvenile characteristic of micropropagated plants [43,58], which may persist in the long term and may facilitate rapid establishment of micropropagated highbush blueberry plants in a new planting area. According to El-Shiekh et al. [62], the enhanced branching and spreading characteristics of tissue culture‑derived blueberry plants can persist for up to 10 years. Scientific reports indicate that plants obtained by micropropagation manifested juvenile features, including high rooting capacity of shoots and intensive vegetative growth [43,58]. Research performed on Vaccinium sp. plants has validated these phenomena in highbush and lowbush blueberries [15, 43-48,56], lingonberries [63,64] and cranberries [65,66].

Reviewer 2:

 “Performed research showed that TC-Ax plants characterized more significant differences to SC plants, than TC-Ad. Perhaps the researchers' focus on finding evidence of somaclonal variability of plants obtained from adventitious shoots,…..” – unclear;

Answer:

To better understand the meaning of the above sentence some modifications have been made:

(line 384-397 The conducted research revealed that plants derived from tissue cultures and propagated by axillary shoots (TC-Ax plants) exhibited more significant differences to conventionally propagated plants (SC) than those propagated by adventitious shoots (TC-Ad). This observation is intriguing, as adventitious shoot cultures are commonly considered to be a source of somaclonal variation. Perhaps the researchers' focus on finding evidence of somaclonal variability in plants obtained from adventitious shoots, generally considered as a source of somaclonal variability, was misplaced. Debnath [79], in his molecular analyses of plants obtained from adventitious shoots, did not include plants obtained from axillary shoots (commonly regarded to be genetically stable). Soneji et al. [80], on the other hand, observed phenotypic differences in pineapple micropropagation using axillary buds, particularly in fruit color and thorn generation [80]. According to the aforementioned authors, cultures derived from axillary shoots are the source of somaclonal variability. The reason for differences between plants propagated in in vitro cultures may be attributed to changes occurring at the genetic and/ or epigenetic level.

Reviewer 2:

77)      Materials and Methods:

4.1. “For the TC-Ax in vitro culture, a 2-node fragment of the shoot, strongly linked with donor culture..” – unclear, what means strongly linked? Further: “weakly linked..”?

Answer:

To better explain the difference between axillary or adventitious shoots derived in vitro culture some sentence was enriched:

(line 455-460)

Two types of in vitro cultures were established from 2-node shoot explants of different origins. For the TC-Ax in vitro culture, a 2-node axillary shoot fragment, strongly connected with the donor culture (by vascular bundles), was used as an explant. Axillary shoots developed from a bud formed in the leaf axils. For the TC-Ad in vitro culture, a 2-node adventitious shoot fragment that had developed from a callus or was weakly connected with the donor culture (without vascular bundles) was used as an explant

Reviewer 2:

88)      Conclusions:

“The background of somaclonal variation may be the result of DNA methylation changes, detected among an analyzed group of plants. It can be supposed that the differences at the morphological, physiological, and biochemical levels, detected among the highbush blueberry plants propagated in different ways, may be the result of differences in the DNA methylation level.” – repetition;

Answer:

The sentence was changed to avoid repetitions:

(line 600-606) The underlying factor for somaclonal variation could be attributed to changes in DNA methylation levels, detected in the analysed plant groups. It can be hypothesised that the detected differences between conventionally and in vitro propagated highbush blueberry plants may be the result of differences in DNA methylation levels. Nevertheless, further analyses should be performed to confirm the present results for other varieties of highbush blueberry plants.

Reviewer 3 Report

Comments and Suggestions for Authors

The review of “Complex analysis of differences between highbush blueberry plants and fruits as an effect of propagation” for IJMS MDPI. The topic of manuscript fits within the scope of the journal IJMS MDPI.

The submitted manuscript of scientific interest. It is well written, has a good structure, consistent experiments and interesting conclusions. Nevertheless, several problems/doubts should be solved before the manuscript is suitable to be published.

Major comments:

The major comment is related to the "Introduction", "Materials and Methods" and "Discussion". In general, the Introduction in this manuscript is somewhat weak. In the introduction after line 36 there is not enough information about biologically active substances in highbush blueberries. In the "Materials and methods" section, information on plant cultivation conditions should be supplemented. In the "Discussion" section, information about the growth regulator 2iP and how it affects plants needs to be added. I also believe that the quality and informativeness of the figures needs to be improved throughout the manuscript.

Specific comments:

L. 28-29. Please check the keywords. The text contains numbers 1, 2, 3, 4.

L. 36-42. I propose to expand this section of the introduction. What other polyphenols (such as flavonoids and phenolic acids) are found in berries besides anthocyanins?

L. 126. Please correct throughout the manuscript: p ≤ 0.05. This note applies to the entire manuscript.

L. 142. Please improve the quality of the figures in the manuscript. Enlarge the image, currently the pictures are very small. Add error bars to represent the standard deviation. Add the name of the ordinate axis (OY) and specify the units of measurement. This note applies to the entire manuscript.

L. 173-174. Check that the units of measurement are correct. In materials and methods mg TE/100g is indicated, and in the figure mg TE/g. Be sure to indicate whether the material was dry or fresh for analysis.

L. 205-207. What do the letters A, B, C mean in Figure 6? Please indicate the transcript in the caption to the Figure 6.

L. 259. Please correct the link to the figure (Figure 2).

L. 325-329. This proposal appears unfinished. Is it known how 2iP affects plants and the content of secondary metabolites (phenolics and flavonoids)? I think this information will strengthen the discussion of the results.

L. 401-410. This section of materials and methods needs to be expanded. How long did the cultivation take place? How often were transplanted to fresh medium? What were the cultivation conditions - lighting, PPFD, photoperiod?

L. 502-505. In what units is the content of ascorbic acid expressed? Check the correct units of measurement in Figure 5.

Author Response

Thank you very much for your thorough analysis of the manuscript originally titled 'Complex analysis of differences between highbush blueberries plants and fruits as an effect of propagation'. 
All the suggestions You provided helped to improve the manuscript. I appreciate Yours comments, both positive and constructive. 
According to comments, regarding the linguistic quality of the text, the manuscript was linguistically proofread by a professional scientific translator, Dr. Grzegorz BurzyÅ„ski. I attached certificates confirming dr. BurzuÅ„ski's competencies. The original title of the manuscript was changed to 'Comprehensive analysis of tallbush blueberry plants propagated in vitro and conventionally. Following Your suggestions, the introduction section of the Manuscript has been largely remodeled and the quality of the figures has been improved. 
Any changes; new text, and content are marked in red. 
I hope that the submitted manuscript will meet with Yours acceptance.

Reviewer 3

In general, the Introduction in this manuscript is somewhat weak.

Answer:

Including the above suggestion and suggestions of other Reviewers, the introduction was rearranged and divided into 5 sections:

1) Information about the blueberries (line 34-50)

2) Methods of propagation (line 51-62)

3) Advantages and disadvantages of propagation methods (line 63-104)

4) MSAP technique (according suegstion of the Reviewer 4th) (line 105-121)

5) The aim of the study (line 122-136)

Reviewer 3

In the introduction after line 36 there is not enough information about biologically active substances in highbush blueberries.

Answer:

The information about biological active substances was included:

(line: 40-43)

The main group of phenolic compounds in blueberries are flavonoids. This group includes anthocyanins, proanthocyanidins, as well as flavonols (mainly quercetin derivatives). The present phenolic acids mainly include chlorogenic, coumaric and elagic acids [3,4].

Reviewer 3

In the "Materials and methods" section, information on plant cultivation conditions should be supplemented.

Answer:

The information about cultivation conditions were included:

(line 460-464) The highbush blueberries cultures were maintained for 10-12 weeks and then passaged to fresh ZB medium. Four passages were performed. Plant cultures were maintained under a 16-h photoperiod, with light provided by cool-white fluorescent lamps (photosynthetic photon flux density [PPF] of approximately 12 µmol m−2 s−1) at 24± 2°C.

Reviewer 3

 In the "Discussion" section, information about the growth regulator 2iP and how it affects plants needs to be added.

Answer:

The information was included as belove:

(line 356-369) Hormone concentration is the major factor in secondary product accumulation such as phenolics and flavonoids [75-77]. Plant hormones are chemical compounds and a group of key signal molecules that are actively involved in the synthesis of plant secondary metabolites and also in regulating development and plant growth [75-77]. According to Baskaran et al. [74], a combination of glutamine and N6-benzyladenine significantly increased the accumulation of phenolics and flavonoids in vitro compared to the separate application of these compounds. In our work, only cytokinins, specifically 2iP (6-γ,γ- dimethylallylamine) at a concentration 10 mg/l, were used in the medium. Research performed by Al-Khayri et al. [75] indicated that cell suspension cultures of date palm (Phoenix dactylifera L.) Palms containing 2,4-D and 2iP yielded the maximum accumulation of phenolics only in the case when 2iP is in combined with 2,5-D in concentration 2.5 mg/L (2iP) and  5 mg/L (2,4-D). Whereas the cell suspension culture medium supplemented with a higher concentration of auxin/cytokinin (10 mg/L 2,4-D + 5 mg/L 2iP) led to accumulated the least concentration of the total phenolic content, and flavonoids of date palm.

Reviewer 3

I also believe that the quality and informativeness of the figures needs to be improved throughout the manuscript.

 Answer:

The quality of the figures has been improved.

Specific comments:

Reviewer 3

  1. 28-29. Please check the keywords. The text contains numbers 1, 2, 3, 4.

 Answer:

It was corrected

 Reviewer 3

  1. 36-42. I propose to expand this section of the introduction. What other polyphenols (such as flavonoids and phenolic acids) are found in berries besides anthocyanins?

Answer:

The information about biological active substances was included:

(line: 40-43)

The main group of phenolic compounds in blueberries are flavonoids. This group includes anthocyanins, proanthocyanidins, as well as flavonols (mainly quercetin derivatives). The present phenolic acids mainly include chlorogenic, coumaric and elagic acids [3,4].

 Reviewer 3

  1. 126. Please correct throughout the manuscript: p≤ 0.05. This note applies to the entire manuscript.

Answer:

It was corrected

 Reviewer 3

  1. 142. Please improve the quality of the figures in the manuscript. Enlarge the image, currently the pictures are very small. Add error bars to represent the standard deviation. Add the name of the ordinate axis (OY) and specify the units of measurement. This note applies to the entire manuscript.

Answer:

All Figures in the manuscript were corrected, including quality, standard deviation, and the names of Y axis with the specific units.

 Reviewer 3

  1. 173-174. Check that the units of measurement are correct. In materials and methods mg TE/100g is indicated, and in the figure mg TE/g. Be sure to indicate whether the material was dry or fresh for analysis.

Answer:

According to the above suggestion, the units in materials and method were verified and corrected. Th proper units. For The antiradical activity the units is [mg TE/g FW] (line 558). For the ascorbic acid  mg/ 100 g FW (line 562-564)

 Reviewer 3

  1. 205-207. What do the letters A, B, C mean in Figure 6? Please indicate the transcript in the caption to the Figure 6.

Answer:

To explain the sign ABC under Figure 6 the underlying text was included: (line 233-234) Differences in DNA bands patterns – products of selective amplification as part of MSAP analysis. Red arrows indicate the lack of specific DNA bands for: in vitro derived plants (TC-Ax and TC-Ad) (A); or for TC-Ax plants only (B,C).

Reviewer 3

  1. 259. Please correct the link to the figure (Figure 2).

Answer:

It was coreccted acording to line 188: Figure 2. Chlorophyll content. Means value of chlorophyll marked with a different lowercase letter ,differ significantly at p ≤ 0.05.

 Reviewer 3

  1. 325-329. This proposal appears unfinished. Is it known how 2iP affects plants and the content of secondary metabolites (phenolics and flavonoids)? I think this information will strengthen the discussion of the results.

Answer:

The information was included as belove:

(line 356-369) Hormone concentration is the major factor in secondary product accumulation such as phenolics and flavonoids [75-77]. Plant hormones are chemical compounds and a group of key signal molecules that are actively involved in the synthesis of plant secondary metabolites and also in regulating development and plant growth [75-77]. According to Baskaran et al. [74], a combination of glutamine and N6-benzyladenine significantly increased the accumulation of phenolics and flavonoids in vitro compared to the separate application of these compounds. In our work, only cytokinins, specifically 2iP (6-γ,γ- dimethylallylamine) at a concentration 10 mg/l, were used in the medium. Research performed by Al-Khayri et al. [75] indicated that cell suspension cultures of date palm (Phoenix dactylifera L.) Palms containing 2,4-D and 2iP yielded the maximum accumulation of phenolics only in the case when 2iP is in combined with 2,5-D in concentration 2.5 mg/L (2iP) and  5 mg/L (2,4-D). Whereas the cell suspension culture medium supplemented with a higher concentration of auxin/cytokinin (10 mg/L 2,4-D + 5 mg/L 2iP) led to accumulated the least concentration of the total phenolic content, and flavonoids of date palm.

Reviewer 3

  1. 401-410. This section of materials and methods needs to be expanded. How long did the cultivation take place? How often were transplanted to fresh medium? What were the cultivation conditions - lighting, PPFD, photoperiod?

Answer:

The information about cultivation conditions were included:

(line 460-464) The highbush blueberries cultures were maintained for 10-12 weeks and then passaged to fresh ZB medium. Four passages were performed. Plant cultures were maintained under a 16-h photoperiod, with light provided by cool-white fluorescent lamps (photosynthetic photon flux density [PPF] of approximately 12 µmol m−2 s−1) at 24± 2°C.

Reviewer 3

502-505. In what units is the content of ascorbic acid expressed? Check the correct units of measurement in Figure 5.

Answer:

According to the above suggestion, the units in materials and method were verified and corrected. Th proper units. For The antiradical activity the units is [mg TE/g FW] (line 558). For the ascorbic acid  mg/ 100 g FW (line 562-564) and Figure 5. (line 220)

Reviewer 4 Report

Comments and Suggestions for Authors

In MS entitled, Complex analysis of differences between highbush blueberry plants and fruits as an effect of propagation, authors have done extensive work and obtained good results. However, MS must be significantly revised. Please see following comments for further improvement.

·       Revise title. It must be clear and concise.

·       In abstract write full form of MSAP technique.

·       In introduction- write about MSAP technique.

·       L513, what is ‘thesis’?

·       L525, can you write 26.9 instead of 26.900?

·       I have one suggestion regarding short forms used in the MS. For- “in vitro with axillary shoots” authors have used short form- TC-Ax. Here I can understand Ax stands for Axillary shoots but what about TC? Same for other short forms. I know it is okay to use non-conventional short forms. However, for reader it becomes difficult to keep track. If possible, change the short forms.

·       In L502 short form of ascorbic acid was mentioned as “VC”. It must be used at the time of first-time use.

·       Revise statement in L502-503. Sentence looks incomplete.

·       Mention the wavelength and standard used to estimate ascorbic acid content.

·       While writing molecular formula of any compound, take care of subscripts and superscripts.

·       In results, data representation must be significantly improved.

·       In Table 1 values presented in wrong manner. 39,4 should be 39.4 and probability value must be less than or equal to 0.05. Check whole MS for this correction.

·       The Means marked with a different lowercase letter must be written as superscript. It will improve the representation.

·       In all the Histograms error bars and titles of Y axis are missing.

·       I am not able to understand Figure 2. I think the Figure must be divided in to Figure 2a and 2b.

·       The figure legends must be elaborated.

·       In Figure 2 and 3, SC should be written in uniform format.

·       In Figure 6,  DNA ladder is missing.

·       Add representative pictures of plants used for analysis to support the quantitative results.

Comments on the Quality of English Language

Extensive language editing is required. 

Author Response

Thank you very much for your thorough analysis of the manuscript originally titled 'Complex analysis of differences between highbush blueberries plants and fruits as an effect of propagation'. 
All the suggestions You provided helped to improve the manuscript. I appreciate Yours comments, both positive and constructive. 
According to comments, regarding the linguistic quality of the text, the manuscript was linguistically proofread by a professional scientific translator, Dr. Grzegorz BurzyÅ„ski.The original title of the manuscript was changed to 'Comprehensive analysis of tallbush blueberry plants propagated in vitro and conventionally. Following Your suggestions, the introduction section of the Manuscript has been largely remodeled and the quality of the figures has been improved. 
Any changes; new text, and content are marked in red. 
I hope that the submitted manuscript will meet with Yours acceptance.

Reviewer 4

  • Revise title. It must be clear and concise.

Answer:

The title was changed to : Comprehensive analysis of highbush blueberry plants propa-gated in vitro and conventionally

Reviewer 4

  • In abstract write full form of MSAP technique.

Answer:

It was included

Reviewer 4

  • In introduction- write about MSAP technique.

Answer:

According to teh sugestion the information about MSAP technique was included

(line 105-121) To identify true-to-type plants obtained after in vitro propagation, it is essential to use methods for evaluating somaclonal variation [49]. A growing number of studies indicate that the variability observed in plants regenerated through in vitro cultures is not solely attributed to genetic alteration but also to epigenetic changes. Additionally, not all genetic variations in somaclones are phenotypically expressed. These modifications can either occur in non-coding sequences may not significantly alter the gene product [49]. To detect somaclonal variation at the epigenetic level, characterized by alterations in DNA methylation, the methylation-sensitive amplification length polymorphism technique (MSAP) is most commonly applied [49-54]. This method is based on the different sensitivity of restriction enzymes to cytosine methylation at their cleavage sites [50, 55]. It permits the comparison of the DNA methylation status of different organisms, based on the differential digestion patterns. A frequently employed method involves HpaII and MspI isoschizomers, both recognizing the same 5′-CCGG sequence. Although HpaII cleaves hemimethylated sequences (only one DNA strand is methylated), it is most sensitive when one or both cytosines are fully methylated (both strands are methylated). In contrast, MspI cleaves at the C5mCGG site, whether hemimethylated or with both strands methylated but does not cleave 5mC5mCGG or 5mCCGG sites [54]

Reviewer 4

 L513, what is ‘thesis’?

Answer:

It was changet to ‘group”

(line 572-573) For each group of plants, 3 pools of berries were extracted yielding 3 biological replicates.

  • L525, can you write 26.9 instead of 26.900?

Answer:

It was corrected

Reviewer 4

  I have one suggestion regarding short forms used in the MS. For- “in vitro with axillary shoots” authors have used short form- TC-Ax. Here I can understand Ax stands for Axillary shoots but what about TC? Same for other short forms. I know it is okay to use non-conventional short forms. However, for reader it becomes difficult to keep track. If possible, change the short forms.

Answer:

The designations TC-Ax and TC-Ad were introduced, according to the use of the abbreviations TC in the literature. TC is used to determine plants originating from in vitro cultures (Tissue Culture derived plants). However, the research reports did not focus on the origin of the explant used to multiplication, but only on comparing plants propagated by in vitro method (TC) or conventionally. In the presented research Plants TC was divided into two groups (TC-Ax and TC-Ad). I reserved the use of the abbreviations AX and AD only for analyses performed on plant cultures. Some of the performed results of in vitro culture with the designation AX and AD were made public at conferences. Therefore, I would like to maintain the terms/abbreviations that were introduced from the beginning based on other scientific reports.

Reviewer 4

  • In L502 short form of ascorbic acid was mentioned as “VC”. It must be used at the time of first-time use.

Answer: We decided to remove the short form VC and used only the whole name of Ascorbic acid

Reviewer 4

Revise statement in L502-503. Sentence looks incomplete.

Answer:  It was corrected according to line 553-558:

The sample for determining antiradical activity (AA) was prepared by homogenising the tissue (5 g) with 15 ml of 75% methanol solution and centrifuging the homogenate at 10,000 x g for 30 min.

The antiradical activity (AA) of fruits was assayed using synthetic ABTS•+ and DPPH• radicals, following the method described by Piechowiak et al. [3]. The results of AA were expressed as Trolox equivalent [mg TE/g FW].

Reviewer 4

  • While writing molecular formula of any compound, take care of subscripts and superscripts.

Answer: 

It was included : acid/H2O (line 567), Na2CO3 (line 576)

Reviewer 4

  • In results, data representation must be significantly improved.

Answer: 

It was included

Reviewer 4

  • In Table 1 values presented in wrong manner. 39,4 should be 39.4 and probability value must be less than or equal to 0.05. Check whole MS for this correction.

Answer:  The commas in all Tables were replaced by a dot.

    The sentence  Included in all the figures : different lowercase letter differ significantly at p ≤ 0.05.

Reviewer 4

The Means marked with a different lowercase letter must be written as superscript (PL: napiany u góry ). It will improve the representation.

Answer:  Above suggestion was included in Table 1 (line 153)

Reviewer 4

  • In all the Histograms error bars and titles of Y axis are missing.

Answer It was included in all Figures 1-5..

Reviewer 4

    I am not able to understand Figure 2. I think the Figure must be divided in to Figure 2a and 2b.

Answer: The The quality of Figure 1 has been improved and the information in figure 1 should be more readable and understandable

Reviewer 4

  • The figure legends must be elaborated.

According to the above sugestion some of the legends were elaborated:

Figure 1. Fluorescence parameters. Means marked with a different lowercase letter differ significantly at p ≤ 0.05.

Figure 2. Chlorophyll content. Means value of chlorophyll marked with a different lowercase letter , differ

significantly at p ≤ 0.05

Figure 3. Results of antioxidant activity. Means marked with a different lowercase letter differ significantly at p ≤ 0.05.

Figure 4. Total anthocyanin and total polyphenol contents. Means marked with a different lowercase letter differ significantly at p<0.05.

Figure 5. Ascorbic acid content. Means marked with a different lowercase letter differ significantly at p ≤ 0.05.

Figure 6. Differences in DNA bands patterns – products of selective amplification as part of MSAP analysis. Red arrows indicate the lack of specific DNA bands for: in vitro derived plants (TC-Ax and TC-Ad) (A); or for TC-Ax plants only (B,C).

Reviewer 4

  • Answer:In Figure 2 and 3, SC should be written in uniform format.

The suggestion was included

Reviewer 4

In Figure 6,  DNA ladder is missing.

  • Answer:Figure 6 shows only polyacrylamide gel fragments, including polymorphic DNA fragments. Due to the poor quality large gel (including 48 wells) Figure 6 does not show the entire polyacrylamide gel.  A marker of the DNA size (ladder), for gel was used at intervals of approximately every 10 wells, so not every separation path had such a marker. Moreover, the above image is only intended to show the occurrence of polymorphic sites and in this case, the size of the products is not that important.

Reviewer 4

  • Add representative pictures of plants used for analysis to support the quantitative results.
  • Answer:

The representative picture of plants and in vitro culture were attached as supplementary

Reviewer 4

Extensive language editing is required. 

  • Answer:

According to suggestions about poor English language and extensive revision needed, the manuscript was revisioned and corrected by professional translation expert Grzegorz Burzyński. Mr Burzyński is a Freelance translator specializing in English-Polish and Polish-English translations of biological, medical, and pharmaceutical texts. He is providing proofreading services for the Journal of Animal and Feed Sciences (JAFS)

Correction confirmation and certificates have been attached as separate documents.

Round 2

Reviewer 1 Report

Comments and Suggestions for Authors

Thanks for your great effort to improve the MS

Best regards

Author Response

Dear Sir and Madam, 

Thank You for taking the time to read the submitted manuscript entitled  ''Comprehensive analysis of highbush blueberry plants propagated in vitro and conventionally (originally entitled 'Complex analysis of differences between highbush blueberry plants and fruits as an effect of propagation'). Thank You for accepting the revised version of the manuscript.

Yours faithfully

Marzena Mazurek

Reviewer 2 Report

Comments and Suggestions for Authors

The suggested improvements have been incorporated into the manuscript. Congratulations on  Authors for their excellent work.

Author Response

(The authors gave the same response as above.)

Reviewer 3 Report

Comments and Suggestions for Authors

I thank the authors for correcting the manuscript.

However, minor corrections still need to be made.

I recommend that authors pay attention to the text in the figures. Some figures have red underlining of the text. Please insert the drawings into the manuscript as a graphic object, for example, an image in vector format (Windows metafile or *.eps) or as a raster image (*.png or *.tif).

Pay attention to lines (204-205), where there is an inscription "p<0.05". Replace with the correct inscription: p ≤ 0.05.

Author Response

Dear Sir and Madam, 

Thank You for taking the time to read the submitted manuscript entitled  ''Comprehensive analysis of highbush blueberry plants propagated in vitro and conventionally (originally entitled 'Complex analysis of differences between highbush blueberry plants and fruits as an effect of propagation'). Thank You for accepting the revised version of the manuscript.

According to suggestion to pay more attention of Figures we  have improved the Figures (captions were corrected without including red underlines), and  insert in Manuscript Figures as a graphic object (jpg files).

We corected also correct inscription: p ≤ 0.05.  as sugested.

Thank You for Your valuable tips to improve  manuscript.

Yours faithfully

Marzena Mazurek

Reviewer 4 Report

Comments and Suggestions for Authors

Authors have significantly improved the MS and the MS can be encouraged for publication.

Comments on the Quality of English Language

Proof reading for language-related issues is required. 

Author Response

Dear Sir and Madam, 

Thank You for taking the time to read the submitted manuscript entitled  ''Comprehensive analysis of highbush blueberry plants propagated in vitro and conventionally (originally entitled 'Complex analysis of differences between highbush blueberry plants and fruits as an effect of propagation'). Thank You for accepting the revised version of the manuscript.  According to sugesstion the  manuscript was proofread for language-related issues. Some language mistakes were corrected (like elagic acids). The text was unified in terms of the 'somaclonal variation', and the term 'in vitro’ was written in italics. References and their occurrence in the manuscript were checked

A few new corrections have been made in yellow color, also the tracking changes possibility was included.

I hope that the version of the submitted manuscript after correction, followed by Yours suggestion, will be fully accepted by You.

Thank You for Your valuable tips to improve  manuscript.
Yours faithfully
Marzena Mazurek